# Preparation of Biomass Activated Carbon Supported Nanoscale Zero-Valent Iron (Nzvi) and Its Application in Decolorization of Methyl Orange from Aqueous Solution

**Bo Zhang * and Daping Wang**

School of Metallurgical and Material Engineering, Hunan University of Technology, Taisshan Road 88, Zhuzhou 412007, Hunan, China

* Correspondence: 13747@hut.edu.cn

**Abstract:** The nanoscale zero-valent iron (nZVI) has great potential to degrade organic polluted wastewater. In this study, the nZVI particles were obtained by the pulse electrodeposition and were loaded on the biomass activated carbon (BC) for synthesizing the composite material of BC-nZVI. The composite material was characterized by SEM-EDS and XRD and was also used for the decolorization of methyl orange (MO) test. The results showed that the 97.94% removal percentage demonstrated its promise in the remediation of dye wastewater for 60 min. The rate of MO matched well with the pseudo-second-order model, and the rate-limiting step may be a chemical sorption between the MO and BC-nZVI. The removal percentage of MO can be effectively improved with higher temperature, larger BC-nZVI dosage, and lower initial concentration of MO at the pH of 7 condition.

**Keywords:** biomass activated carbon; methyl orange; pulse electrodeposition; zero valent iron nanoparticles

## 1. Introduction

Dye has become a widespread environmental pollution problem because of its wide application in industry such as paper, textiles, plastic, and leather tanning industries in the past several decades [1,2]. Over 100,000 commercial dyes are associated with an annual production rate of over 800,000 tons [3]. Large amounts of dye-containing effluents pose great challenges to the environment because of their strong color, complex structure, stability, and low biodegradability [4]. Therefore, it is of vital importance to remove dyes from wastewater to protect the aquatic life and alleviate the crisis of water pollution.

Many methods such as adsorption [5], reduction [6], advanced oxidation processes [7], coagulation [8], membrane separation [9], and biological methods [10] are used to remove dyes from wastewater. In recent years, nanoscale zero-valent iron (nZVI) particle has been extensively used as a new tool for the treatment of wastewater contaminated with various pollutants as a result of its small particle size, large specific surface area, and high reactivity [11]. Unfortunately, there are still some technical challenges in the use of nZVI. On the one hand, because of interparticle Van der Waals and magnetic interactions, nZVI particles are prone to agglomeration, resulting in the significant decrease of their dispersibility [12]. On the other hand, nanoparticles tend to be oxidized, and the formation of oxide layers easily block the serviceable active surface sites, which finally diminishes the reactivity [13]. To get rid of these shortcomings, supporting nZVI particles is an option. For the past few years, most of previous studies have loaded nZVI particles on some porous materials as a

carrier, such as kaolinite [14], bentonite [15], resin [16], activated carbon [17], mesoporous carbon [18], mesoporous silica [19], titanium oxide [20], pumice [1,21], and grapheme [22], through the liquid phase reduction method. Nevertheless, there still has some problems associated with method, such as low production efficiency, high production cost, and large amounts of hydrogen the generation during the preparation process [23,24]. All of these problems lead to the limitations of large-scale application.

In this study, the compound material was prepared with two steps. First, the nZVI particles were directly obtained from steel scrap using the pulse electrodeposition. Second, the nZVI particles were quickly supported to the biomass activated carbon (BC) under the mechanical agitation condition. The obtained BC-nZVI material was characterized and its decontamination abilities were tested by the removal of methyl orange (MO). In addition, the effects of dosage, initial pH, initial concentration, and temperature on the removal percentage of MO were investigated in conjunction with the analyses of mechanism and kinetics. It is hoped that this study can provide a new preparation technique for nZVI composites for wastewater treatment.

## 2. Materials and Methods

### 2.1. Materials and Chemicals

The biomass activated carbon was carbonized from coconut shell with particle size of 2~4 mm, filling density of 0.5~0.55 g/mL, PH value of 6.5~7.5, and iodine adsorption value of 850~1000 mg/g. This was supplied by Lu-yuan Co. Ltd., Mianyang, China. Ferrous sulfate septihydrate (FeSO$_4$·7H$_2$O, ≥98.5%), methyl orange (MO, 99%), sodium hydroxide (NaOH, 98%), hydrochloric acid (HCl, 36%), sodium dodecyl benzene sulfonate (DBS, 99%), thiourea (≥98%), and absolute ethanol (99%) were obtained from Sinopharm Chemical Reagent Co. Ltd., Shanghai, China. All the chemicals were analytical reagent grade and deionized water from a Liceng UPA-L system (18.2 MΩ/cm, 25 °C) was used for all experiments.

### 2.2. Preparation of nZVI and Composite with Biomass Activated Carbon

FeSO$_4$·7H$_2$O was dissolved in deionized water to prepare an electrolyte with Fe$^{2+}$ of 30 g/L. The additive (thiourea, 0.5 g/L and DBS, 1.0 g/L) was dispersed and emulsified in deionized water in an ultrasonic cleaner (VGT SONIC-L30 300 w, 28 kHz) and added to the electrolyte. The electrolysis was carried out at 50~60 °C and the anode plate for electrolysis was a waste carbon steel (Q235) plate, which was used after surface treated. The cathode plate for electrolysis was a stainless-steel box and several ultrasonic vibrators were added in the box. The ultrasonic power was 28 KHz. The cathode and the anode were separated by a filter membrane and the structure of the electrolytic cell is shown in Figure 1. After 10 h of electrolysis, the electrolyte near the cathode plate was rapidly pumped out and filtration, and the leached residue was washed with deionized water and absolute ethanol for three times, respectively. Subsequently, DBS and absolute ethanol were intermixed with leached residue and dispersed in a mechanical disperser (FS400D, 2000 rpm, Qiwei instrument Co. Ltd., Hangzhou, China). Finally, an emulsion of nZVI was prepared after 12 h of dispersion under nitrogen protective.

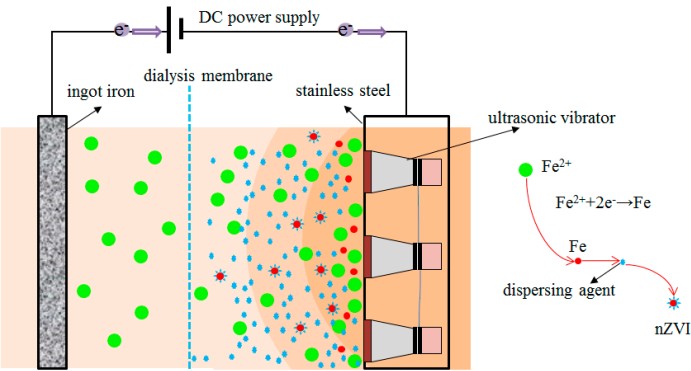

**Figure 1.** Electrolytic cell for preparation of nanoscale zero-valent iron (nZVI).

The electrolytic process occurs at the cathode and anode as follows [25]:
Anode:

$$Fe - 2e = Fe^{2+} \tag{1}$$

Cathode:

$$Fe^{2+} + 2e = Fe \tag{2}$$

The iron particles generated on the cathode plate were quickly stripped and emulsified in the electrolyte under the action of cavitation effect of ultrasonic oscillation and combined with dispersant. The polymer molecular chain of dispersant was negatively charged, which can form a brush surface layer to hinder the aggregation. Under the action of dispersant, iron particles were difficult to agglomerate and oxidize.

The emulsion of nZVI was transferred to a conical flask (1000 mL) for 100 mL, and 200 mL deionized water and 200 g biomass activated carbon were added into conical flask. Then, the conical flask was sealed with a sealing plug and shocked on a horizontal vibrator (HY-5A) with a frequency of 240 rpm. After 10 h of shocking, the mixture was cleaned three times by absolute ethanol to remove the excess emulsion. Finally, the freshly composite material BC-nZVI was dried at 30 °C in the vacuum drying oven for 12 h and kept in a nitrogen atmosphere prior to use.

### 2.3. Characterization and Analytic Methods

The surface morphology images of BC-nZVI were obtained with a scanning electron microscope (SEM) (SIGMA 300; Carl Zeiss AG, Oberkochen, Hallbergmoos, Germany) operating at 30 kV. The crystal structure and composition phase was analyzed by X-ray diffraction (XRD) (Philips-X'Pert Pro MPD, Malvern Panalytical, Almelo, Holland, The Netherlands). The sample of BC-nZVI was dissolved by HCl and the total iron ions in acid solution were analyzed by inductively Coupled plasma spectrometer (ICP-MS 8000; Perkinelmer, Waltham, MA,　USA) to determine the actual load of nZVI on BC. The concentration of MO solution was measured using a UV-Spectrophotometer (760CRT, Shanghai precision scientific instrument co., LTD, Shanghai, China) at $\lambda_{max} = 464$ nm [26].

### 2.4. Batch Experiments

The removal percentage of MO by BC-nZVI was evaluated by batch experiments. The degradation test for MO was carried out in a 250 mL conical flask and the volume of the experimental solution was 150 mL. The flask was placed in the horizontal vibrator and wobbled at 240 rpm after the BC-nZVI was added. The initial pH value of solution was adjusted by 0.01 mol/L HCl and 0.01 mol/L NaOH. The five major factors (MO initial concentration (20–200 mg/L), weight of BC-nZVI dosage (0.25–0.75 mg/L), reaction temperature (20°C~80°C), pH (3.0~10.0), and interaction time (60–120 min)) were considered for optimizing MO degradation by BC-nZVI. The samples (4 mL) were collected within a specified time and the concentration of MO was estimated using UV-

Spectrophotometer after centrifuged. In order to ensure the quality of the data, all experiments were conducted in three copies and the average value was reported.

The MO removal percentage was calculated by Equation (3):

$$R(\%) = \frac{(C_0 - C_e) \times 100}{C_0} \tag{3}$$

The equilibrium removal capacity of MO ($q_e$ (mg/g)) was calculated by Equation (4):

$$q_e = \frac{(C_0 - C_e) \times V}{W} \tag{4}$$

where $C_0$ (mg/L) is initial concentration of MO, $C_e$ (mg/L) is the equilibrium concentrations of MO, $V$ (L) is the experimental solution volume, and W (g) is dry weight of nZVI in BC-nZVI used.

## 3. Results and Discussions

### 3.1. Characterization of nZVI and BC-nZVI

The morphologies of nZVI and BC-nZVI were determined using SEM and presented in Figure 2. Figure 2a shows that the nZVI nanoparticles were substantially nearly smooth and spherical with sizes ranging from 40 to 80 nm. The nanoparticles were slightly agglomerated, and this may have occurred during the detection process when the emulsion of nZVI was placed on the observation table and rapidly dried by washing ear ball. The surface of BC presents a porous structure, which provides a good platform for loading the nZVI as shown in Figure 2b. The nanoparticles were substantially evenly distributed on the biomass activated carbon surface and pores under mechanical loading condition.

The XRD patterns of nZVI, BC, and BC-nZVI were obtained and presented in Figure 3. No characteristic diffraction peaks of Fe were observed because of its weak crystallization and the inclusion of DBS. The characteristic peaks have several small peaks at $2\theta$ = 17.41°, 19.24°, and 3.84° on nZVI wave lines, which represent iron oxide [22], indicating a certain oxidation in the nanoparticles. In the characteristic peak of the BC-nZVI wave lines, there were obvious carbon peaks at $2\theta$ = 26.60°, 43.45°, 54.79° [23] and the iron oxide peak disappears, indicating that the loading was beneficial to prevent the oxidation of the nZVI.

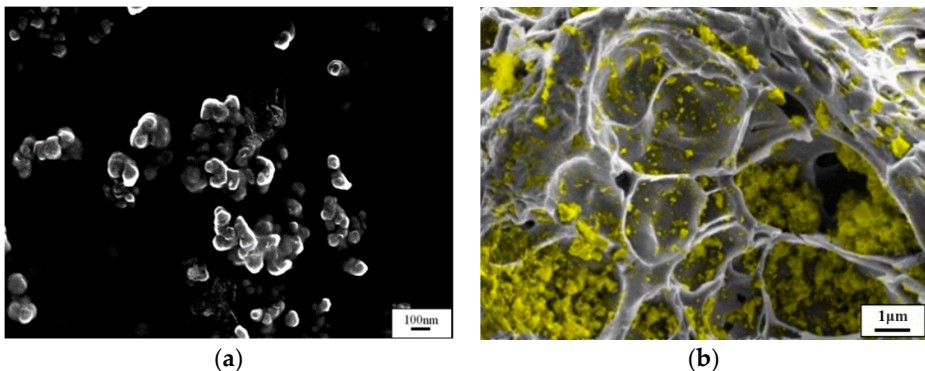

(**a**)  (**b**)

**Figure 2.** SEM images of (**a**) nZVI, (**b**) biomass activated carbon (BC)-nZVI.

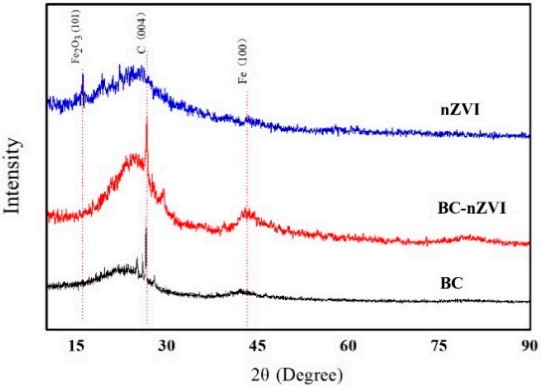

**Figure 3.** XRD pattern of nZVI, BC, and BC-nZVI.

### 3.2. Removal Efficiencies of MO

The removal percentage of MO was investigated using BC, nZVI, and BC-nZVI, respectively, as shown in Figure 4. Obviously, the total removal percentage of BC-nZVI and nZVI are significantly higher than that of BC as a result of the contribution of nZVI. But for nZVI and BC-nZVI, the different tendencies were presented. In the early stage (within 20 min), nZVI holds the higher removal percentage than that of BC-nZVI. The reason behind this fact is that the nZVI was in direct contact with the aqueous solution and the reaction was fast, while the removal percentage of BC-nZVI is lower because the pore structure of BC prevents the contact bewteen nZVI and MO. However, the opposite results can be found after 20 min, and the BC-nZVI shows the higher removal percentage than that of nZVI. This is due to the fact that the sole nZVI particle was easily oxidized than BC-nZVI composite material, resulting in the decreased reaction activity.

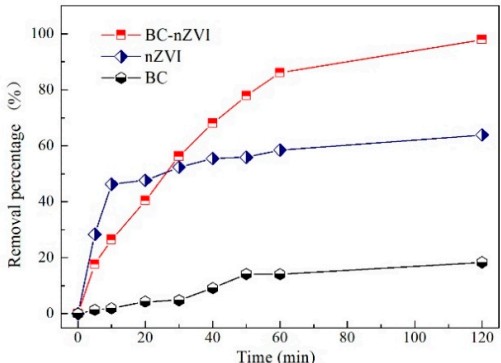

**Figure 4.** Comparative degradation of MO using different materials. The dosage of BC, nZVI, and BC-nZVI were 3 g/L, 0.5 g/L, and 0.5 g/L (the dry weight of nZVI in BC-nZVI used) with an initial MO concentration of 200 mg/L, temperature of 25 °C, and original pH.

Figure 5 showed the UV–vis absorption spectra of the MO before and after the addition of BC-nZVI. For the original MO, there were two main absorption bands at 464 and 270 nm, which were attributed to a conjugate structure formed by the azo band under the effects of the electronic-donation of dimethylamino group and the $\pi-\pi$ * transition of aromatic rings [27]. Obviously, the band intensity of 464 and 270 nm decreased with reaction time, and almost disappeared after reaction 50 min. It proves the degradation of MO through the cracking of azo bonds. The XRD pattern of BC-nZVI after interaction with MO was represented in Figure 6. The characteristic peaks of 27.87°, 50.17° present the iron (III) oxide $Fe_2O_3$. With the increase of reaction time, the intensity of $Fe_2O_3$ characteristic peaks increased. The radicals of H were generated by the reaction nZVI nanoparticles and $H_2O$ or hydrogen ion [28,29], which caused the azo bond to open and consequently the absorption bands at 464 nm and 270 nm vanished.

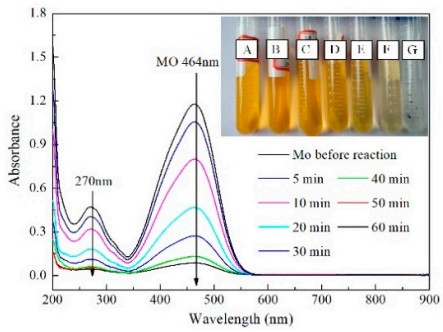

**Figure 5.** UV–vis patterns of degradation of MO.

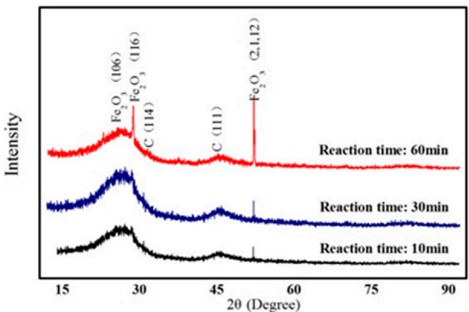

**Figure 6.** XRD pattern of BC-nZVI after reaction.

The maximum removal capacities of BC-nZVI in this study were compared with those of other absorbents prepared by liquid phase reduction method, as shown in Table 1. Clearly, the present adsorbent (BC-nZVI) shows the same level of removal capacity as other efficient absorbents, such as B-nZVI, HJ-NZVI, nZVI/BC, and Bio-nZVI. However, it should be emphasized that the BC-nZVI in this study was synthesized by pulse electrodeposition and mechanical agitation method, and this method is easier and more cost-effective than the liquid phase reduction method.

**Table 1.** Comparison of MO removal with different absorbents reported in literature.

| Absorbent | Dosage | Initial pH | Initial Concentration | T °C | Reaction Time | Removal Capacity | Refs. |
|---|---|---|---|---|---|---|---|
| B-nZVI | 0.5 g/L | Or | 200 mg/L | 30 | 60 min | 93.75% | Chen et al. 2011 [13] |
| HJ-NZVI5 | 1.0 g/L | Or | 200 mg/L | 25 | 45 min | 72.8% | Li et al. 2017 [30] |
| nZVI/BC | 0.6 g/L | Or | 200 mg/L | NA | 60 min | 96.17% | Yang et al. 2018 [31] |
| B */nZVI/Pd | 0.5 g/L | Or | 200 mg/L | 27 | 90 min | 91.87% | Wang et al. 2013 [32] |
| nZVI/Pd | 0.5 g/L | Or | 200 mg/L | 27 | 90 min | 85% | Wang et al. 2013 [32] |
| BC-nZVI | 0.5 g/L | Or | 200 mg/L | 25 | 120 min | 97.94% | This study |
| nZVI | 0.5 g/L | Or | 200 mg/L | 25 | 120 min | 63.9% | This study |

B: synthesized bentonite; HJ: Hangjin clay; BC: biochar; Bio: Biogenic; B *: Functional clay; Or is 6.17; NA: not available

## 4. Impact of Operational Parameters

### 4.1. Effect of Dosage

As illustrated in Figure 7a, the decolorization kinetics of MO was dependent on the BC-nZVI dosage. The larger the BC-nZVI dosage, the higher the removal percentage. The concentration of MO decreases dramatically in the initial 20 min at the highest BC-nZVI dosage of 0.75 g/L, while the removal percentage at 0.25 g/L was sluggish. Moreover, for the dosages of 0.25, 0.5, and 0.75 g/L, the

maximum removal percentages of MO were 68.96%, 94.41%, and 97.43%, and the removal capacities of BC-nZVI were 344.1 mg/g, 300.8 mg/g, and 181.2 mg/g, respectively. Due to the loading effect of BC, the oxidation of nZVI was slowed down, resulting in the highest removal percentage.

When the amount of BC-nZVI dosage was insufficient, the slower kinetics and lower removal percentage were observed for MO removal, but the removal capacity of BC-nZVI was higher [32]. At a sufficient amount of BC-nZVI, enough nZVI activity sites was provided to react with MO in the initial stage, so the reaction kinetics were faster, but the removal capacity of BC-nZVI was lower.

### 4.2. Effect of pH

The effects of initial pH on MO removal were illustrated in Figure 7b. With the increase of pH from 3.0 to 10.0, the removal percentage of MO was declined from 97.75% to 89.13%, and decreased from 76.01% to 21.08% in the first 10 min. It suggests that the lower pH value was beneficial to MO reduction by BC-nZVI. The possible explanation is that the more $H^+$ was released at lower pH values, which could accelerate the corrosion of nZVI particles, and eliminate ferrous hydroxide on the surface of nZVI particles to generate fresh active sites. The removal capacities of BC-nZVI were 193.5 mg/g, 195.4 mg/g, and 180.1 mg/g as the pH were 3.0, 7.0, and 10.0, respectively. The alkaline pH reduced the removal capacity of BC-nZVI because $OH^-$ would significantly enhance the formation of the iron hydroxide, which formed a surface layer on the nZVI particles and inhibited further reactions [20].

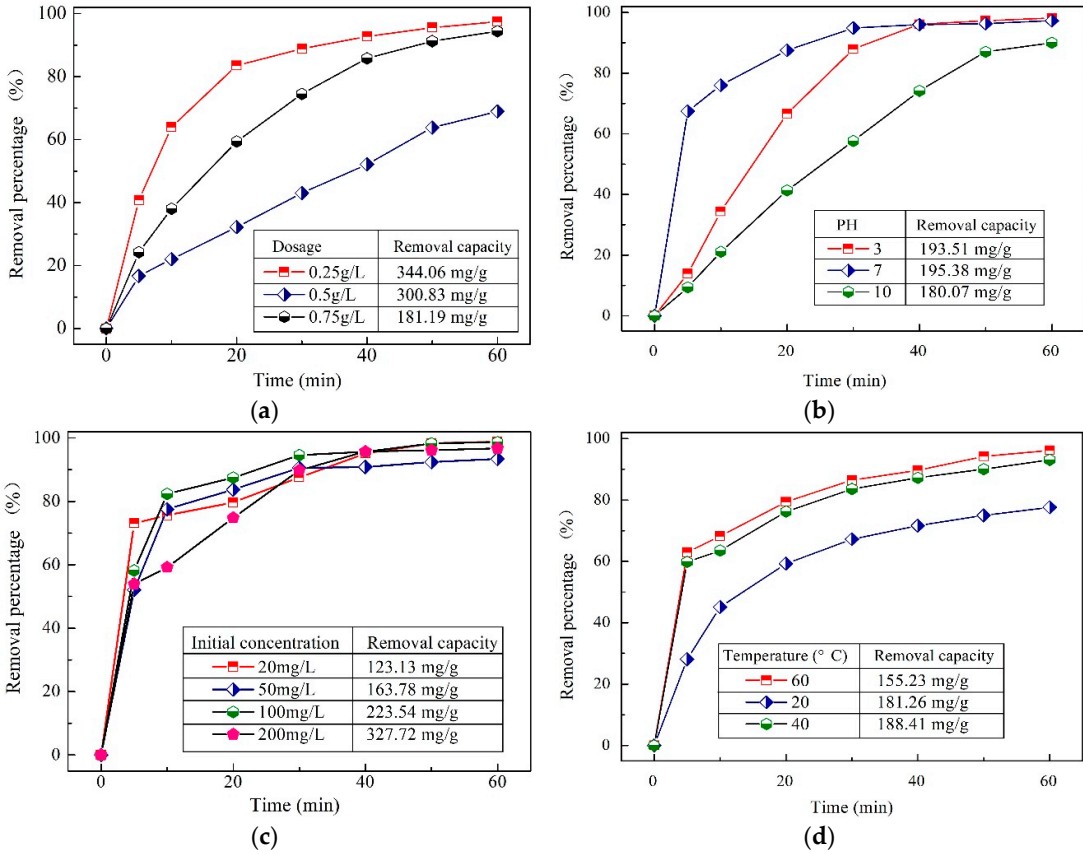

**Figure 7.** (**a**) Effect of dosage on degradation of MO, with an initial MO concentration of 200 mg/L, temperature of 25 °C, and original pH; (**b**) Effect of the pH values on degradation of MO, in the context of 0.5 g/L BC-nZVI with an initial MO concentration of 50 mg/L; (**c**) Effect of initial concentration on degradation of MO, at the original pH using 0.5 g/L BC-nZVI; (**d**) Effect of temperature on degradation of MO, in the context of 0.5 g/L BC-nZVI with an initial MO concentration of 50 mg/L, temperature of 25 °C, and the original pH.

### 4.3. Effect of Initial Concentration

　　　The effects of initial MO concentrations ranging from 20 to 200 mg/L on removal percentage was shown in Figure 7c. The maximum decolorization efficiencies were 98.8%, 98.7%, 96.7%, and 93.3%, and the decolorization efficiencies in first 20 min were 77.6%, 82.3%, 75.6%, and 59.2% at initial concentrations of 20 mg/L, 50 mg/L, 100 mg/L, and 200 mg/L, respectively. The removal percentage with the initial concentration of 200 mg/L was significantly lower than that of the other lower initial concentration. It may be caused by that the highly initial MO concentration leads to a competition effect among the MO molecules and a decline in decolorization efficiency [33,34]. However, with the increase of initial concentration, the removal capacity of nZVI was obviously improved. At a higher initial concentration, the removal capacity of nZVI was significant higher, at 327.7 mg/g.

### 4.4. Effect of Temperature

　　　As shown in Figure 7d, the removal percentage of MO increased with increasing temperature. The final removal percentages of MO were 77.6%, 93.1%, and 96.2%, respectively, for 20, 40, and 60 °C. Similar results were reported previously [35–37]. There may be two possible explanations for this phenomenon: (1) The mobility of MO was increased by higher temperature, and (2) the activation energy of decolorizing reaction was increased as the temperature rose.

### 4.5. Kinetics of Degradation of MO

　　　In order to investigate the mechanism of degradation, the pseudo-first-order kinetics model (PFO) (Equation (5)) and the pseudo-second-order model (PSO) (Equation (6)) were generally used to test the degradation of MO using BC-nZVI, which can be expressed as the following equation [13,29]:

$$\ln(q_e - q_t) = \ln q_e - k_1 t \tag{5}$$

$$t/q_1 = 1/(k_2 q_e^2) + t/q_e \tag{6}$$

where $q_e$ and $q_t$ (mg/g) are the amount of MO adsorbed per gram of BC-nZVI at equilibrium and at time $t$, respectively, while $k_1$ (min$^{-1}$) and $k_2$ (g/(mg·min)) are the PFO and PSO rate constants, respectively.

　　　The effect of initial MO concentrations on the removal efficiency was calculated with these two kinetic models, and the fitting of the kinetics data was shown in Figure 8 and Table 2.

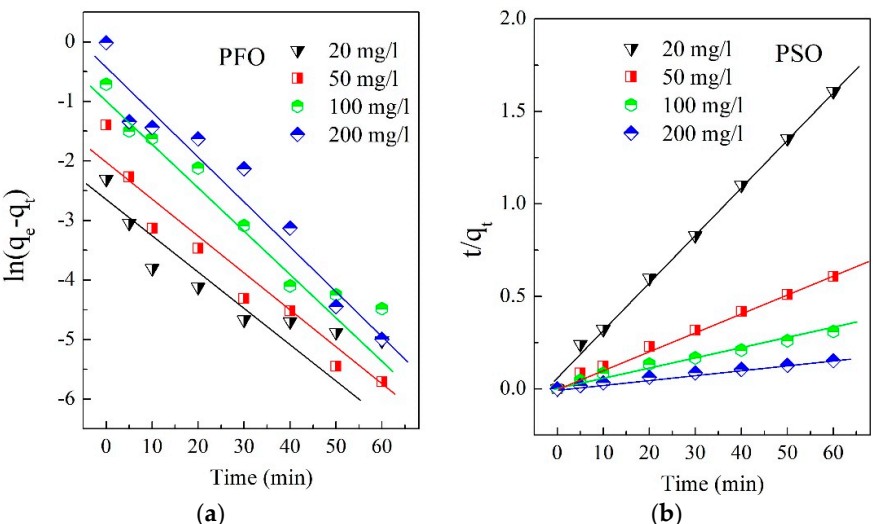

**Figure 8.** Kinetic modeling of MO adsorption using pseudo-first-order (PFO) (**a**) and pseudo-secondo-rder (PSO) model (**b**).

**Table 2.** Kinetic parameters for MO removal.

| C$_0$ (mg/l) | q$_e$ (mg/g) | PFO | | | PSO | | |
|---|---|---|---|---|---|---|---|
| | | q$_e$, cal | k$_1$ | R$^2$ | q$_e$, cal | K$_2$ | R$^2$ |
| 20 | 123.13 | 121.93 | 0.0749 | 0.8231 | 124.29 | 0.0259 | 0.997 |
| 50 | 163.78 | 161.77 | 0.0647 | 0.9484 | 163.50 | 0.0098 | 0.9974 |
| 100 | 223.54 | 211.06 | 0.067 | 0.9597 | 229.42 | 0.0048 | 0.9879 |
| 200 | 327.72 | 319.11 | 0.0401 | 0.9466 | 329.98 | 0.0025 | 0.9933 |

Significantly, the pseudo-second-order model matched better with the data ($R^2$ = 0.988) than the pseudo-first-order model ($R^2$ = 0.823), suggesting that the chemical sorption may be the main rate-limiting step between the MO and BC-nZVI.

The rate constants for a pseudo second-order reaction were 0.0259 g/(mg·min) for 20 mg/L, 0.0048 g/(mg·min) for 100 mg/L, and 0.0025 g/(mg·min) for 200 mg/L, respectively. The result indicates that the degradation of MO occurs in the interface of BC-nZVI [20,22], hence the rate of degradation was closely linked to the initial concentration of MO and the active surface sites of BC-nZVI, as discussed in the previous section. It should be emphasized that the BC-nZVI materials in our study were prepared by the mechanical agitation method and most of nZVI particles are mainly distributed in the outer layer of BC. It means the MO could easily contact with the nZVI particles, so the chemical sorption is the main rate-limiting step compared to the mass transfer process. However, some other X-nZVI materials synthesized by liquid phase reduction method agreed with the pseudo-first-order model [14–18]. Here, X presents the carrier, such as clay [15], activated carbon [17], and mesoporous carbon [18]. This is due to that the nZVI particles can be uniform distributed in both the outer layer and internal space of carriers (X) using the liquid phase reduction method. Therefore, the mass transfer may be the main limiting step under this condition.

## 5. Conclusions

The present study demonstrated that BC-nZVI particles can be used to efficiently degrade MO in aqueous solution by cleaving the azo linkages. SEM, XRD, UV–vis, and batch experiments confirmed the characteristics of this composite: (1) The nZVI nanoparticles prepare dby means of pulse electrodeposition were substantially nearly smooth and spherical with sizes ranging from 40 to 80 nm, and successfully loaded on the surface and inside the pores of BC; (2) BC as a dispersant and stabilizer, which reduced the extent of aggregation of nZVI and therefore enhanced reactivity; (3) compared with other adsorbents prepared by liquid phase reduction method, BC-nZVI reflected with the same maximum removal rate; and (4) batch experiments show that various parameters such as dosage, pH, initial concentration of MO, and temperature did affect the removal of MO. In addition, the BC-nZVI in this study was synthesized by pulse electrodeposition and mechanical agitation method, and this method is easier and more cost-effective than the liquid phase reduction method, and the 97.94% removal percentage demonstrated its promise in the remediation of dye wastewater.

**Author Contributions:** B. Z. designed and carried out this research. D. P. W. prepared and analyzed the data. B.-Z. wrote this paper. All authors have approved the manuscript.

**Funding:** This study was funded by the Specialized Research Fund for the National Natural Science Foundation of China (No. 51504090), and the Natural Science Foundation of Hunan Province, China (No. 2019JJ60062).

**Acknowledgments:** The authors are grateful to the three anonymous reviewers for their insightful and constructive comments, which greatly improved the quality of the paper.

**Conflicts of Interest:** The authors declare no conflict of interest.

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
