# Peer review of "Preparation of Biomass Activated Carbon Supported Nanoscale Zero-Valent Iron (Nzvi) and Its Application in Decolorization of Methyl Orange from Aqueous Solution"

_water, doi:10.3390/w11081671_

Round 1

Reviewer 1 Report

In recent years, the use of zero-valent iron (ZVI) for the treatment of toxic contaminants in groundwater has received wide attention.

In order to overcome a series of defects in the practical application of ordinary ZVI, the present study focused on the synthesis of nanoscale zero-valent iron (nZVI) particles, suppported on biomass activated carbon (BC), in order to obtain the composite material BC-nZVI.

The characterization of this material was carried out by electron microscope, X-ray diffraction and measure of total iron to determine the load of nZVI on BC. Finally, the removal efficiency of BC-nZVI toward methyl orange (MO) was tested. The degradation efficiency was evaluated by varying several parameters, such as pH, temperature, BC-nZVI dosage, MO concentration.

The topic of the manuscript is suitable for publication on “Water”.

Despite the scarce novelty of this work, due to the high number of studies focused on the degradation of methyl orange by nanoscale zero-valent iron (supported or not), the Authors presented a great amount of data: however, some elaboration based on experimental results could be added, such as the study of the process kinetic.

In addition, Figures 7 a-d are very difficult to understand: each figure reports two different graphs (a x/y dispersion and a histogram) and four axes. For the sake of readable charts, I suggest to “split” each figure into two separated graphs or one graph+one table.

Finally, the mechanism of interaction and reaction between MO and BC-nZVI should be added and discussed, before reporting the experimental results.

In the light of the above, I would suggest accepting this paper after some MAJOR REVISIONS have been made.

In the following, some particular comments.

L 160. “shown” should be replaced by “showed”

L 165. “Mo” should be replaced by “MO”

Table 1. It is very interesting that the Authors compare their results with literature data, but the conditions (MO initial concentration, reaction time, dosage) are very different each other, making the removal efficiencies difficult to compare

L 196. “PH” should be replaced by “pH”

Figure 7 (in addition to the advice of splitting each figure, see above)The removal efficiency has been evaluated by varying BC-nZVI dosage (fig 7a), pH (fig 7b) and temperature (fig 7d) at different MO concentration (200, 0.5 and 50 mg/l respectively). In my opinion, the MO concentrations should be the same or, at least, similar.

LL 212-217. I’m sorry but the general meaning is not clear: LL 212-215 seem in contradiction with LL 215-217. Please, rewrite these sentences in order to clarify the influence of MO concentration on the removal efficiency.

Reviewer 2 Report

no comments

Author Response

Thanks your review

Reviewer 3 Report

1. General Comments:

This paper investigated the "activated carbon supported zero-valent iron in removing methyl orange from aqueous solution".

Clean and safe drinking water is essential for human beings. However, tap water can be contaminated by several contaminations such as organic contaminants (Liu et al., 2019). One of the most promising ways to remove organic pollutants is by using adsorbents (Mojiri et al., 2019).

Liu, X., Lai, D., Wang, Y., 2019. J Hazard Mater., 361, 37-48. doi: 10.1016/j.jhazmat.2018.08.082

Mojiri, A., Kazeroon, R.A., Gholami, A., 2019. Water, 11(3), 551. doi: 10.3390/w11030551

2. Abstract:

2.1. Page 1, Line 14; "The results showed that the composite material of BC-nZVI exhibits the excellent performance..." Give the value of removal efficiency.

2.2. Page 1, Line 16; "In addition, the effects of different operational parameters, such as the BC-nZVI dosage, pH value of aqueous solution, initial MO concentration and test temperature, on the decolorization process of MO were investigated...." Just investigated!!!! I could not understand the effects of each parameters on the removal efficiency by reading the abstract!! What about the performance? You should mention to the values or results.

2.3. Write keywords alphabetically!

3. Introduction:

3.1. Page 1, Lines 40-44; "Taking the clay-supported nZVI as an example. The clay was placed in a three-neck open flask, with an iron solution added and stirred for 20 minutes. Then, the freshly prepared NaBH4 solution was slowly added to the mixture and stirred for 1 hours. Lastly, the composite material was dried in a vacuum after excess NaBH4 removed. Nano-iron particles can be uniformly distribute" It is not suitable for introduction. Besides, It is not related to your paper!

3.2. The introduction should be completely re-written! Strats from problems and issues related to organic pollutants in aqueous solutions, such as methyl orange and dye. Then talk about biomass activated carbon and nZVI. Afterwards, bring up gap of the knowledge and problem statement by using previous study. What is the novelty and contribution of the current study? 

4. Materials and Methods:

4.1. Page 2, Line 60; "The biomass activated carbon was carbonized from coconut shell with particle size of...." Give details about preparing activated carbon. How? Which method? Details should be described well.

4.2. Page 2, Line 68; "Preparation of nZVI and composite with biomass activated carbon" Support this part with references!

4.3. Page 2, Line 75; "and the structure of the electrolytic cell was shown..." It should be edited to "and the structure of the electrolytic cell is shown..."  

5. Results and Discussion:

5.1. The discussion part should be improved. Compare your results with previous studies in details. 

5.2. Adsorption isotherm studies should be done. 

Round 2

Reviewer 3 Report

It seems that the reviewers’ comments have been addressed well. It can be published in the current version.